# REAL-TIME AUTOML

## ABSTRACT

We present a new zero-shot approach to automated machine learning (AutoML) that predicts a high-quality model for a supervised learning task and dataset in real-time without fitting a single model. In contrast, most AutoML systems require tens or hundreds of model evaluations. Hence our approach accelerates AutoML by orders of magnitude. Our method uses a transformer-based language embedding to represent datasets and algorithms using their free-text descriptions and a meta-feature extractor to represent the data. We train a graph neural network in which each node represents a dataset to predict the best machine learning pipeline for a new test dataset. The graph neural network generalizes to new datasets and new sets of datasets. Our approach leverages the progress of unsupervised representation learning in natural language processing to provide a significant boost to AutoML. Performance is competitive with state-of-the-art AutoML systems while reducing running time from minutes to seconds and prediction time from minutes to milliseconds, providing AutoML in real-time.

## 1 INTRODUCTION

A data scientist facing a challenging new supervised learning task does not generally invent a new algorithm. Instead, they consider what they know about the dataset and which algorithms have worked well for similar datasets in past experience. Automated machine learning (AutoML) seeks to automate these tasks to enable widespread use of machine learning by non-experts. A major challenge is to develop *fast, efficient* algorithms to accelerate applications of machine learning (Kokiopoulou et al., 2019). This work develops automated solutions that exploit human expertise to learn *which datasets are similar* and *what algorithms perform best*. We use a transformer-based language model (Devlin et al., 2018) allowing our AutoML system to process text descriptions of datasets and algorithms, and a feature extractor (BYU-DML, 2019) to represent the data itself. Using such models for our representation brings in large-scale data. We allow to train our model on other existing AutoML system solutions, specifically AutoSklearn (Feurer et al., 2015), AlphaD3M (Drori et al., 2018), OBOE (Yang et al., 2019), and TPOT (Olson & Moore, 2019), tapping into their diverse set of solutions. Our approach fuses these representations (dataset description, data, AutoML pipeline descriptions) and represents datasets as nodes in a graph of datasets.

Generally, graph neural networks are used for three main tasks: (i) node prediction, (ii) link prediction, and (iii) sub-graph or entire graph classification. In this work we use a GNN for node prediction, which predicts the machine learning pipeline for an unseen dataset. Specifically, we use a graph attention network (GAT) (Veličković et al., 2018) with neighborhood aggregation, in which an attention function adaptively controls the contribution of neighbors. An advantage of using a GNN for AutoML is boosting AutoML performance by sharing information between datasets (graph nodes): including description and algorithm, by message passing between the nodes in the graph. In addition, GNNs generalize well to a new unknown dataset using the aggregated weights learnt over the training datasets. GNN weights are shared with the test dataset for prediction. GNNs generalize to entire new sets of datasets. Finally, prediction is in real-time, within milliseconds.

A simple idea is to use machine learning pipelines that performed well (for the same task) on similar datasets. What constitutes a similar dataset? The success of an AutoML system often hinges on this question, and different frameworks have different answers: for example, AutoSklearn (Feurer et al., 2015) computes a set of *meta-features*, which are features describing the data features, for each dataset, while OBOE (Yang et al., 2019) uses the performance of a few fast, informative models to compute latent features. More generally, for any supervised learning task, one can view the list of

recommended algorithms generated by any AutoML system as a vector describing that task. This work is the first to use the information that a human would check first: a summary description of the dataset and algorithms, written in free text. These dataset features induce a metric structure on the space of datasets. Under an ideal metric, a model that performs well on one dataset would also perform well on nearby datasets. The methods we develop in this work show how to learn such a metric using the recommendations of an AutoML framework together with the dataset description. We provide a new zero-shot AutoML method that predicts accurate machine learning pipelines for an unseen dataset and classification task in real-time and runs the pipeline in a few seconds.

We use a transformer-based language model to embed the description of the dataset and pipelines and a feature extractor to compute meta-features from the data. Based on the description embedding and meta-features, we build a graph as the input to a graph neural network (GNN). Each dataset represents a node in the graph, together with its corresponding feature vector. The GNN is trained to predict a machine learning pipeline for a new node (dataset). Therefore, given a new dataset, our real-time AutoML method predicts a pipeline with good performance within milliseconds. The running time of our predicted pipeline is a few seconds and the accuracy of the predicted pipeline is competitive with state-of-the-art AutoML methods that are given one minute. This work makes several contributions by using language embeddings and GNNs for AutoML for the first time, and leveraging existing AutoML systems. The result is a real-time high-quality AutoML system.

**Real-time.** Our system predicts a machine learning pipeline for a new dataset in milliseconds and then runs the pipeline and tunes its hyper-parameters within three seconds. This reduces computation time by orders of magnitude compared with state-of-the-art AutoML systems, while improving performance.

**GNN architecture.** Our work achieves real-time AutoML by introducing several architectural components that are new to AutoML. These include embeddings for *dataset descriptions* and *algorithm descriptions* using a state-of-the-art transformer-based language model in addition to (standard) embeddings for data; a non-Euclidean embedding of datasets as a graph; and a predictive model employing a GNN on the graph of datasets. Importantly, the GNN recommends a pipeline for a new dataset by adding a node to the graph of datasets and sharing the GNN weights with the new node. Using the information and relationships between all datasets boosts AutoML performance.

**Embeddings.** Bringing techniques from NLP to AutoML, specifically using a large-scale transformer-based language model to embed the description of the dataset and algorithms, brings in information from a large corpra of text. This allows our zero-shot AutoML to train on a small set of datasets with state-of-the-art test set performance.

**Leveraging existing AutoML systems.** Our flexible architecture can use pipeline recommendations from any number of other AutoML systems to improve performance.

## 2 RELATED WORK

AutoML is an emerging field of machine learning with the potential to transform the practice of Data Science by automatically choosing a model to best fit the data. Several comprehensive surveys of the field are available (He et al., 2019; Zöller & Huber, 2019).

**Processing each dataset in isolation.** The most straightforward approach to AutoML considers each dataset in isolation and asks how to choose the best hyper-parameter settings for a given algorithm. While the most popular method is still grid search, other more efficient approaches include Bayesian optimization (Snoek et al., 2012) or random search (Solis & Wets, 1981).

**Recommender systems.** These methods learn (often, exhaustively) what algorithms and hyper-parameter settings performed best for a training set-of-datasets and use this information to select better algorithms on a test set without exhaustive search. This approach reduces the time required to find a good model. An example is OBOE (Yang et al., 2019; 2020), which fits a low rank model to learn the low-dimensional representations for the models (or pipelines) and datasets that best predict the cross-validated errors, among all bilinear models. To find promising models for a new dataset,

OBOE runs a set of fast but informative algorithms on the new dataset and uses their cross-validated errors to infer the feature vector for the new dataset. A related approach (Fusi et al., 2018) using probabilistic matrix factorization powers Microsoft Azure's AutoML service (Mukunthu, 2019).

**Search trees.** Auto-Tuned Models (Swearingen et al., 2017) represent the search space as a tree with nodes being algorithms or hyper-parameters and searches for the best branch using a multi-armed bandit.

**Model-based reinforcement learning.** AlphaD3M (Drori et al., 2018; 2019a) formulated AutoML as a single player game. The system uses reinforcement learning with self-play and a pre-trained model which generalizes from many different datasets and similar tasks.

**Genetic programming.** TPOT (Olson & Moore, 2019) and Autostacker (Chen et al., 2018) use genetic programming to choose both hyper-parameter settings and a topology of a machine learning pipeline. TPOT represents pipelines as trees, whereas Autostacker represents them as layers.

**Bayesian optimization.** AutoSklearn (Feurer et al., 2015) chooses a model for a new dataset by first computing (ad hoc) data meta-features to find nearest-neighbor datasets. The best-performing methods on the neighbors are refined via Bayesian optimization and used to form an ensemble.

**Differentiable programming.** End-to-end learning of machine learning pipelines is performed using differentiable primitives (Milutinovic et al., 2017) forming a directed acyclic graph.

**Algorithmic primitives.** One major factor in the performance of an AutoML system is the base set of algorithms it can use to compose more complex pipelines. For a fair comparison, in our numerical experiments we compare our proposed methods only to other AutoML systems that use Scikit-learn (Pedregosa et al., 2011) primitives.

**Embeddings.** Language has a common unstructured representation as a sequence of words, sentences, or paragraphs. The most significant recent progress in NLP is large-scale transformer-based models and embeddings (Devlin et al., 2018; Shoeybi et al., 2019; Raffel et al., 2019) based on attention mechanisms (Vaswani et al., 2017). An unsupervised corpus of text is transformed into a supervised dataset by defining content-target pairs along the entire text: for example, target words that appear in each sentence, or target sentences which appear in each paragraph. A language model is first trained to learn a low dimensional embedding of words or sentences followed by a map from low dimensional content to target (Mikolov et al., 2013). This embedding is then used on a new, unseen and small dataset in the same low-dimensional space. Our work uses such embeddings for automatic machine learning. In a similar fashion to recent work (Drori et al., 2019b) we use an embedding for the dataset and algorithm descriptions. In this work we model the non-linear interactions between these embedding using a neural network as well.

## 3 METHODS

Our zero-shot AutoML predicts a machine learning pipeline for a classification task on a dataset based on the dataset description and data, and based on other datasets, their relationships, and their recommended pipelines by AutoML systems. We embed the dataset description and extract data meta-features to construct a graph of datasets where each node represents a dataset. The graph is processed using a graph neural network (GNN). Each node of the graph contains a feature vector which is the fusion of the description embedding and data meta-features, and the GNN node representations includes other AutoML solutions. The machine learning pipeline for a new dataset is predicted by the GNN. A detailed architecture is illustrated in Figure 1 and described by Algorithms 2 and 3. The notation used in this work are given in Table 1.

### 3.1 PRE-PROCESSING

Our pre-processing consists of (i) dataset description embedding; (ii) dataset meta-feature extraction; and (iii) pipeline computation and description embeddings, as described next and summarized in Algorithm 1.

| Notation | Description |
|---|---|
| $\mathcal{D}$ | Dataset |
| $\mathcal{M}(\mathcal{D})$ | Dataset description |
| $\mathcal{P}$ | Machine learning pipeline |
| $\mathcal{M}(\mathcal{P})$ | Machine learning pipeline description |
| C $\in$ O, S, A, T | OBOE, AutoSklearn, AlphaD3M, TPOT |
| $\mathcal{P}_C(\mathcal{D})$ | Pipeline recommended by C on dataset $\mathcal{D}$ |
| $\mathcal{P}_\star(\mathcal{D})$ | Best pipeline on dataset $\mathcal{D}$ |
| $\hat{\mathcal{P}}(\mathcal{D})$ | Predicted pipeline on dataset $\mathcal{D}$ |
| $\mathcal{R}(\mathcal{P}, \mathcal{D})$ | Performance of running pipeline $\mathcal{P}$ on dataset $\mathcal{D}$ |
| $\mathcal{F}_\mathcal{D}$ | Data meta-features |
| $\mathcal{F}_\mathcal{M} = E(\mathcal{M}(\mathcal{D}))$ | Embedding of dataset description |
| $\mathcal{F}_{\mathcal{D},\mathcal{M}} = [\mathcal{F}_\mathcal{D}, \mathcal{F}_\mathcal{M}]$ | Concatenation |
| $\mathcal{F}_\mathcal{P} = E(\mathcal{M}(\mathcal{P}))$ | Embedding of pipeline description |
| $\mathcal{G}$ | Datasets graph |
| $i \in V$ | Node in $\mathcal{G}$ |
| $j \in \mathcal{N}(i)$ | Neighbors $j$ of node $i$ |
| $\mathcal{F}_i = f_\phi(\mathcal{F}_{\mathcal{D}_i}, \mathcal{M}_i)$ | Fusion network output on graph node |
| $\mathbf{v}_i = [\mathcal{F}_i, \mathcal{F}_{\mathcal{P}_\star(\mathcal{D}_i)}]$ | Features of node in $\mathcal{G}$ |
| $\mathbf{u}_i = g_\theta(\mathbf{v}_i)$ | Fusion network, features of node in GNN |
| $\{\mathbf{u}_j\}_{j \in \mathcal{N}(i)}$ | Features of node neighbors in GNN |
| $h_{W,z}(\mathbf{u}_i, \{\mathbf{u}_j\}_{j \in \mathcal{N}(i)})$ | GNN with parameters $W, z$ |

Table 1: Zero-shot AutoML notation and description.

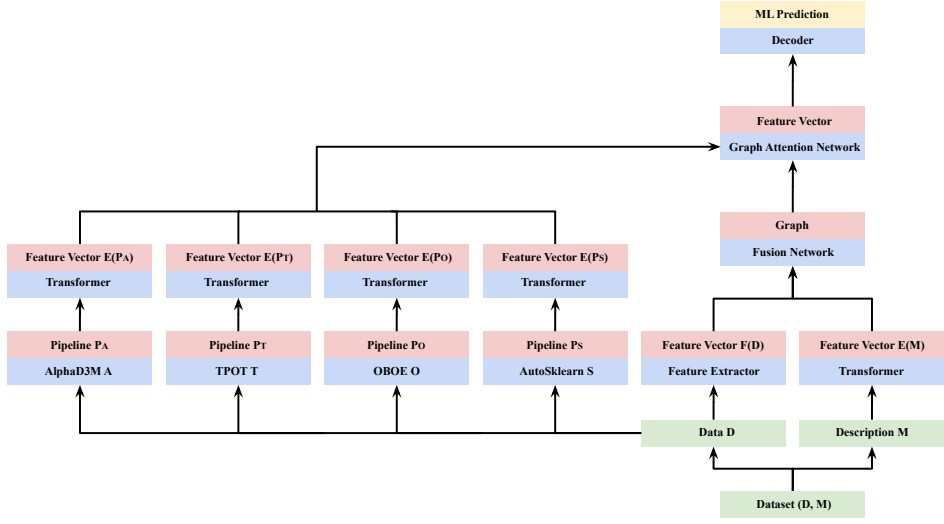

Figure 1: Zero-shot AutoML architecture: Dataset descriptions are embedded using a language model. The data itself is passed through a feature extractor. Other AutoML system algorithms are embedded using a language model. Fully connected neural networks fuse together the encoded feature vectors. A graph captures the relationships between the embedded representations. At training time a GNN learns the aggregation of each node in the graph and its neighbors. The GNN predicts a pipeline for a new node (dataset). At test time a dataset is added as a new node in the graph and the GNN predicts the best machine learning pipeline without running any AutoML system or evaluating any pipeline. Inputs are colored green, neural networks in blue, intermediate outputs in red, and predicted output in yellow.

**Dataset Description Embedding.** We create a feature vector by embeding the description $\mathcal{M}(\mathcal{D})$ of each dataset as a 1024-dimensional vector $\mathcal{F}_\mathcal{M} = E(\mathcal{M}(\mathcal{D})) \in \mathbb{R}^{1024}$ using BERT (Devlin et al., 2018). The supplementary material shows examples of dataset descriptions embedded using our approach.

**Data meta-features.** We compute meta-features $\mathcal{F}_\mathcal{D} \in \mathbb{R}^{148}$ for the dataset $\mathcal{D}$ using a feature extractor (BYU-DML, 2019), restricting to meta-features that can be computed in one second on any of the datasets used in our experiments. Meta-features include statistics of the datasets and results of simple algorithms.

**Pipelines and pipeline embedding.** For each dataset, we compute the recommended pipeline returned by AutoML systems OBOE (O), AutoSklearn (S), AlphaD3M (A), and TPOT (T). We create feature vectors for recommended pipelines by embedding the Scikit-learn documentations for pre-processor or feature selector and estimator (which is unique within each pipeline). Again, we use the BERT embedding, to form a 1024-dimensional embedding $E(\mathcal{M}(\mathcal{P}_C(\mathcal{D}))) \in \mathbb{R}^{1024}$ for each pipeline, where $C$ ranges over the AutoML methods O, S, A, and T. The best-performing pipeline $\mathcal{P}^\star$ returned by any AutoML system serves as our training label: we train our system to recommend this pipeline.

**Fused dataset representations.** The combined representation of dataset $\mathcal{D}_i$ with description $\mathcal{M}(\mathcal{D}_i)$ fuses together the dataset description embedding and data meta-features using a neural network (whose weights are learned):

$$\mathcal{F}_i = f_\phi([\mathcal{F}_{\mathcal{D}_i}, \mathcal{F}_{\mathcal{M}_i}]) \in \mathbb{R}^{512}. \tag{1}$$

We also represent the dataset and its best pipeline by fusing this representation with the pipeline embedding using a second neural network:

$$\mathbf{u}_i = g_\theta([\mathcal{F}_i, \mathcal{F}_{\mathcal{P}_\star(\mathcal{D}_i)}]) \in \mathbb{R}^{512} \tag{2}$$

Experimentally, these fused representations improve performance compared to concatenation.

---

**Algorithm 1** Zero-shot AutoML pre-processing

> **Input:** training datasets $\{(\mathcal{D}_i, \mathcal{M}_i)\}_{i \in V}$.
> **Output:** features $\{\mathcal{F}_{\mathcal{M}i}, \mathcal{F}_{\mathcal{D}i}, \mathcal{F}_{\mathcal{P}_\star(\mathcal{D}_i)}\}_{i \in V}$.
> **for** $i = 1$ **to** $n$ **do**
>   compute embedding of description $\mathcal{F}_{\mathcal{M}i} = E(\mathcal{M}_i)$
>   compute data meta-features $\mathcal{F}_{\mathcal{D}i}$
>   **for all** C $\in$ O, S, A, T **do**
>     compute recommended pipeline $\mathcal{P}_C(\mathcal{D}_i)$
>     compute performance on dataset $\mathcal{R}(\mathcal{P}_C, \mathcal{D}_i)$
>   **end for**
>   select best performing pipeline $\mathcal{P}_\star(\mathcal{D}_i)$
>   embed pipeline $\mathcal{F}_{\mathcal{P}_\star(\mathcal{D}_i)} = E(\mathcal{M}(\mathcal{P}_\star(\mathcal{D}_i)))$
> **end for**

---

## 3.2 GRAPH REPRESENTATION

We build a graph $\mathcal{G} = (V, E)$ where each node $i \in V$ represents the dataset $\mathcal{D}_i$ and has feature vector $\mathbf{v}_i$.

**Nodes.** The feature vector $\mathbf{v}_i = [\mathcal{F}_i, \mathcal{F}_{\mathcal{P}_C(\mathcal{D}_i)}] \in \mathbb{R}^{1536}$ for node $i$ representing dataset $\mathcal{D}_i$ with description $\mathcal{M}(\mathcal{D}_i)$ concatenates the fused dataset representation (described above) $\mathcal{F}_i \in \mathbb{R}^{512}$ and the pipeline embedding $\mathcal{F}_\mathcal{P} = E(\mathcal{P}_\star) \in \mathbb{R}^{1024}$ for the pipeline $\mathcal{P}_\star$ that performed best on the dataset. During training, we mask the pipeline embedding from the feature vector and learn to predict a node using the GNN.

**Edges.** To compute the edges of the graph $\mathcal{G}$, we compute the distance $d$ between each pair of datasets $i, j$ as $d = \|\mathcal{F}_i - \mathcal{F}_j\|_2$ where $\mathcal{F}_i$ and $\mathcal{F}_j$ are the fused dataset representations (described above) for the datasets. Two datasets are connected by an edge if dataset $j$ is one of the $k$ nearest neighbors of dataset $i$ or vice versa. In our experiments, we chose $k = 20$: we found that our method is reasonably robust to the choice of $k$; that Euclidean distance outperforms cosine similarity; and that a k-NN graph outperforms a threshold-based graph.

At training time, we build this graph on the training datasets. At test time, given a new test dataset, we dynamically connect the new node to the graph using its fused feature representation $f_\phi([\mathcal{F}_{\mathcal{M}_{\text{test}}}, \mathcal{F}_{\mathcal{D}_{\text{test}}}])$ to choose edges. Notice that the edges for the new dataset are chosen quickly, without fitting a single machine learning model.

## 3.3 NEURAL NETWORK ARCHITECTURE

The neural networks we train for zero-shot AutoML consist of two fusion networks and a graph attention network (a type of GNN). The fusion networks are used to capture the non-linear inter-

actions between the features corresponding to the dataset description, the data meta-features, and the pipeline embedding. The GNN predicts the best pipeline for a new dataset based the weights optimized during training as described next.

**Graph Attention Network.** A graph attention network (GAT) (Veličković et al., 2018) is used to predict the best pipeline for a new dataset. Each layer $l = 1, ..., L$ of the GNN updates the feature vector at the $i$-th node as:

$$\mathbf{u}_i^l = \alpha_{ii} W \mathbf{u}_i^{l-1} + \sum_{j \in \mathcal{N}(i)} \alpha_{ij} W \mathbf{u}_j^{l-1}, \tag{3}$$

where $W$ is a learnable weight matrix, $\mathcal{N}(i)$ are the neighbors of the $i$-th node, and $\alpha_{ij}$ are the attention coefficients, defined as:

$$\alpha_{ij} = \frac{\exp(\sigma(z^\top [W\mathbf{u}_i, W\mathbf{u}_j]))}{\sum_{k \in \mathcal{N}(i)} \exp(\sigma(z^\top [W\mathbf{u}_i, W\mathbf{u}_k]))}, \tag{4}$$

where $z$ is a learnable vector, and $\sigma(\cdot)$ is the leaky ReLU activation function.

Our GNN consists of 3 GAT layers. The last layer of our GNN is a softmax which computes a vector of probabilities over pipelines. Hence the output of the GAT is a probability distribution over pipelines for each node. The network recommends the pipeline that maximizes this probability. Alternatively, we may sample from this probability distribution to obtain several pipelines that can be combined into an ensemble.

## 3.4 TRAINING AND TESTING

**Training.** Our training process is illustrated in Figures 1 and 2, and described in Algorithm 2. At each training iteration, we randomly select a node $i$. We mask the pipeline embedding of the $i$-th node as $\mathbf{u}_i = g_\theta([\mathcal{F}_i, \mathbf{0}])$. The true label is defined as the pipeline with best performance among the four AutoML systems $\mathcal{P}_\star(\mathcal{D}_i)$ on the $i$-th dataset. The resulting problem is a multi-class classification problem with as many classes as there are distinct algorithms.

The loss function is defined by cross-entropy between the probability $\hat{\mathbf{p}}$ of predicted algorithm $\hat{\mathcal{P}}(\mathcal{D}_i)$ and one-hot encoding $\mathbf{y}$ of the best algorithm $\mathcal{P}_\star(\mathcal{D}_i)$:

$$\mathcal{L}(\hat{\mathcal{P}}(\mathcal{D}_i), \mathcal{P}_\star(\mathcal{D}_i)) = -\sum_{l=1}^{m} \mathbf{y}_l \log(\hat{\mathbf{p}}_l). \tag{5}$$

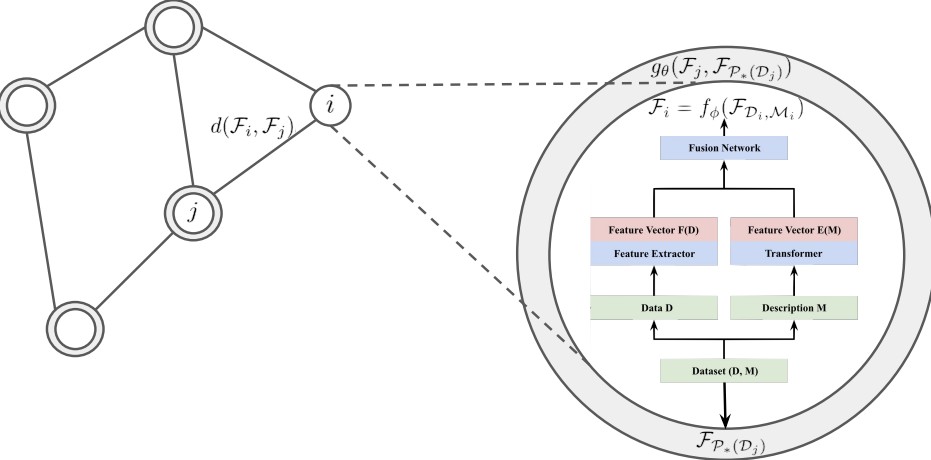

Figure 2: Illustration of zero-shot AutoML dataset graph construction and prediction.

---

**Algorithm 2** Zero-shot AutoML training

> **Input:** training datasets, descriptions $\{\mathcal{D}_i, \mathcal{M}(\mathcal{D}_i)\}_{i \in V}$.
> **Output:** datasets graph $\mathcal{G}$, GNN $h_{W,z}$, fusion networks $f_\phi$ and $g_\theta$.
> pre-process: compute $\{\mathcal{F}_{\mathcal{M}_i}, \mathcal{F}_{\mathcal{D}_i}, \mathcal{F}_{\mathcal{P}_\star(\mathcal{D}_i)}\}_{i \in V}$
> initialize fusion networks weights $\phi, \theta$.
> initialize GNN weights $W, z$.
> **for** each backprop iteration **do**
>     generate updated datasets graph $\mathcal{G}$:
>     **for** $i = 1$ **to** $n$ **do**
>         compute fused representation $\mathcal{F}_i = f_\phi(\mathcal{F}_{\mathcal{D}_i, \mathcal{M}_i})$
>     **end for**
>     compute pairwise distances $d(\mathcal{F}_i, \mathcal{F}_j)_{i,j \in V}$
>     **for** $i = 1$ **to** $n$ **do**
>         connect node $i$ to k-NN nodes $\mathcal{N}(i)$
>     **end for**
>     select random node $i$ in $\mathcal{G}$
>     compute $\mathbf{u}_i = g_\theta(\mathcal{F}_i, \mathbf{0})$
>     **for all** $j \neq i$ **do**
>         compute $\mathbf{u}_j = g_\theta(\mathcal{F}_j, \mathcal{F}_{\mathcal{P}_*(\mathcal{D}_j)})$
>     **end for**
>     predict best pipeline $\hat{\mathcal{P}}(\mathcal{D}_i) = h_{W,z}(\mathbf{u}_i, \{\mathbf{u}_j\}_{j \in \mathcal{N}(i)})$
>     compute loss $\mathcal{L}(\hat{\mathcal{P}}(\mathcal{D}_i), \mathcal{P}_\star(\mathcal{D}_i))$
>     update weights
> **end for**

---

**Testing.** Our testing process is illustrated on the right path of Figure 1 and Algorithm 3. Given a new dataset $\mathcal{D}$ and description $\mathcal{M}$, we compute the description embedding $\mathcal{F}_{\mathcal{M}}$ and data meta-features $\mathcal{F}_{\mathcal{D}}$ and the fused dataset representation $\mathcal{F}$. We use this representation to compute the edges of this new node in the graph of all datasets. Next, we add the new node, with features $\mathbf{u} = g_\theta([\mathcal{F}, \mathbf{0}])$, to the current graph, replacing the embedding of the pipeline with the zero vector. Finally, we use the graph neural network to recommend a pipeline for the test dataset.

---

**Algorithm 3** Zero-shot AutoML testing

> **Input:** dataset $\mathcal{D}_i$, description $\mathcal{M}(\mathcal{D}_i)$, datasets graph $\mathcal{G}$, GNN, s.t. $i \notin V$ (disjoint train and test).
> **Output:** predict best pipeline $\hat{\mathcal{P}}(\mathcal{D}_i)$ for task on dataset.
> generate new node $i$ in $\mathcal{G}$:
> compute embedding of description $\mathcal{F}_{\mathcal{M}} = E(\mathcal{M}(\mathcal{D}_i))$
> compute data meta-features $\mathcal{F}_{\mathcal{D}}$
> compute fused representation $\mathcal{F} = f_\phi(\mathcal{F}_{\mathcal{D}}, \mathcal{F}_{\mathcal{M}})$
> connect node $i$ to k-NN nodes $j \in \mathcal{N}(i)$, $V = V \cup \{i\}$.
> compute $\mathbf{u}_i = g_\theta(\mathcal{F}, \mathbf{0})$
> predict best pipeline $\hat{\mathcal{P}}(\mathcal{D}_i) = h_{W,z}(\mathbf{u}_i, \{\mathbf{u}_j\}_{j \in \mathcal{N}(i)})$

---

Notice our method does not need to complete even a single model fit to recommend a model with hyper-parameters. On the other hand, we must fit the model (to learn the parameters) on the dataset to predict output values for new input data. Our method can always recommend a model in 3 seconds, but training is still needed for prediction.

## 4 RESULTS

Table 2 shows our results for a representative set of test datasets, comparing our approach with state-of-the-art AutoML systems and baselines. For each dataset (row), Table 2 reports the mean evaluation accuracy of different AutoML methods. Figure 3 compares the of accuracy on the test set between our zero-shot approach given 3 seconds of computation, and other AutoML systems and random forest baseline given 1 minute of computation. Our new zero-shot AutoML approach is the only AutoML system that provides predictions within 3 seconds. OBOE requires at least 20 seconds to perform predictions, and then only on a few of the datasets. AlphaD3M reaches performance slightly better than our approach, however given a minute of computation. See the supplementary

|  | | 3 seconds | | | | 1 minute | | |
| Dataset | Zero-Shot | Linear | Random Forest | OBOE | AutoSklearn | TPOT | AlphaD3M | Random Forest |
| --- | --- | --- | --- | --- | --- | --- | --- | --- |
| Lymph | 0.867 | 0.800 | 0.670 | 0.800 | 0.800 | 0.867 | **0.933** | 0.867 |
| Heart-C | 0.774 | 0.710 | 0.806 | 0.742 | 0.806 | 0.742 | **0.871** | 0.839 |
| Vehicle | 0.776 | 0.612 | 0.729 | 0.835 | **0.871** | 0.824 | **0.871** | 0.729 |
| Hayes-Roth | **0.769** | 0.404 | 0.731 | 0.692 | 0.712 | 0.654 | 0.750 | **0.769** |
| Colleges | **0.838** | 0.726 | 0.786 | 0.803 | 0.812 | 0.812 | **0.838** | 0.812 |
| KC1 | 0.872 | 0.848 | 0.829 | 0.251 | 0.872 | 0.877 | **0.886** | 0.872 |
| Banana | 0.745 | 0.551 | 0.881 | 0.808 | 0.898 | **0.911** | **0.911** | 0.900 |
| Cardi | **1.000** | 0.432 | **1.000** | **1.000** | **1.000** | **1.000** | **1.000** | **1.000** |
| Cnae-9 | 0.935 | 0.954 | 0.954 | 0.935 | 0.954 | 0.889 | **0.972** | 0.916 |
| Seeds | 0.952 | 0.905 | 0.905 | 0.952 | 0.952 | **1.000** | **1.000** | 0.905 |
| Wall-Robot | **1.000** | 0.907 | **1.000** | **1.000** | **1.000** | **1.000** | **1.000** | 0.998 |
| Cardi-Multi | **0.995** | 0.869 | 0.986 | **0.995** | 0.986 | **0.995** | **0.995** | **0.995** |
| BachChoral | 0.776 | 0.580 | 0.786 | 0.212 | **0.817** | 0.774 | 0.797 | 0.787 |
| Cjs | 0.982 | 0.846 | 0.971 | 0.978 | **1.000** | **1.000** | **1.000** | 0.925 |
| LED-Display | 0.760 | 0.740 | 0.680 | **0.800** | 0.740 | 0.740 | **0.800** | 0.760 |
| Wine-Quality | 0.686 | 0.451 | 0.678 | 0.484 | **0.982** | 0.686 | 0.714 | 0.692 |
| SpeedDating | 0.851 | **0.870** | 0.847 | 0.837 | 0.865 | 0.835 | **0.870** | 0.862 |
| Mofn | **1.000** | **1.000** | **1.000** | **1.000** | **1.000** | **1.000** | **1.000** | **1.000** |

Table 2: Comparison of testing performance and time between AutoML systems and baselines: our zero-shot approach given 3 seconds and AutoSklearn, OBOE, TPOT, and AlphaD3M given 1 minute. Testing time for predicting the machine learning algorithm is milliseconds. Testing time for running the predicted machine learning algorithm and computing performance is 3 seconds. Our new zero-shot AutoML approach is the only AutoML system that provides predictions within 3 seconds.

material for additional results which validate the performance of our method. First, our zero-shot method generally outperforms other simple baselines given the same amount of computation time. Second, our zero-shot method, in 3 seconds, gives results comparable to state-of-the-art AutoML systems given 1 minute.

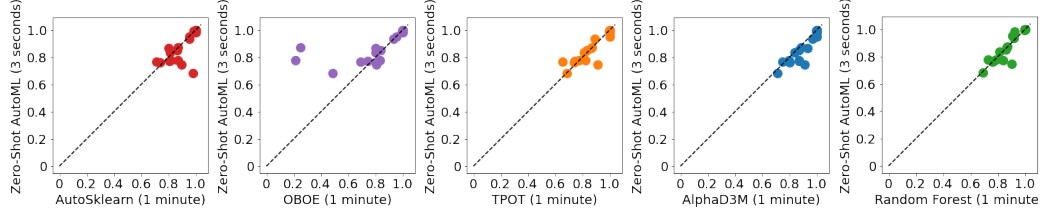

Figure 3: Comparison of accuracy on test set between our zero-shot approach given 3 seconds of computation and other AutoML systems and Random Forest baseline given 1 minute of computation. Our zero-shot approach matches the performance of baselines while running 20 times faster.

## 5 CONCLUSIONS AND FUTURE WORK

We introduce a new zero-shot approach to AutoML that is able to recommend a good pipeline to use for a given dataset in real-time. Our system builds a graph from both NLP text embedding of the dataset and pipeline descriptions as well as data meta-features and uses a graph neural network to predict the best pipeline for a given dataset. Our approach matches the performance of other state-of-the-art AutoML systems and is significantly faster, reducing running time from minutes to seconds and prediction time from minutes to milliseconds. Future work will extend our approach to handle different types of data, including audio and images. In addition, we envision an extension to semi-supervised AutoML by using a GNN to embed a large unsupervised set of datasets without pipelines, such as the 25 Million datasets available on Google dataset search (Brickley et al., 2019), together with a small supervised set of datasets with AutoML pipelines. Finally, we will make our data, models, and code public upon publication.

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
