# OpenReview forum: "Real-Time AutoML"
_ICLR.cc/2021/Conference — Reject_

### Official Review · AnonReviewer3 · 2020-10-27
**We carefully review the motivation, approach, and empirical results.**

**Rating:** 4
**Confidence:** 5

**Review:**

##### Summary
The authors propose an AutoML by combining graph neural network (GNN) and transformer-based embeddings, the recommended pipelines from existing four AutoML systems (alphaD3M, TPOT, OBOE, autoSklearn) into its design. Specifically, the proposed framework is novel because
- it creates the dataset description embedding by BERT, extract meta features of the dataset, and generate data embedding from description embedding and the meta features.
- it will select the best pipeline predicted from the four AutoML frameworks and then convert the pipeline into pipeline embedding.  The best pipeline is used as the ground truth label for training the GNN.
- it creates a graph with datasets as nodes where edges denote similarity between dataset embeddings.
- the dataset graph is used as input to GNN (specifically, Graph Attention Network) to learn the GNN weights
- for a new test dataset, the trained GNN simply predicts a ML pipeline by dynamically adding it as node in the dataset graph via K-nearest neighbors

##### Clarity
The paper is mostly well written and easily understandable even for readers who are new to AutoML. The framework is clearly explained in Section 3. Result section in Section 4 is limited under the page constraint. However, I strongly feel the authors should bring **TABLE 3** and corresponding discussions in **Appendix C: Computation Time** from Supplementary into the main paper especially when the proposed model pitches real-time performance.


##### Main Questions
1. In Introduction,  "....develops automated ***solution**  that ~~**exploit**~~  ***leverage** human expertise...."  The only human component I found the system is using is the textual descriptions provided for the dataset and estimators. The authors should rephrase the statement not to imply human-in-loop system.
2. In Methods,  Figure 1: Please use notations in figure consistent with Table 1.
3. Algorithm 3 (Testing): Is the proposed system scalable since for every new test dataset? Because the proposed method needs to compute K-NN to create new edges.
4. Figure 3: What does each point in the graph denote?
5. What was the performance of other AutoML systems in Table 2 at 3 seconds ? Also, what is the reason for having a 1 minute time limit for these baselines? I understood limit of 3 seconds for zero-shot AutoML since *Cardi* test dataset took ~3 seconds as per Table 3 in Supplementary.
6. What were the ML pipelines predicted by the proposed zero-shot AutoML system and other AutoML systems for the test datasets in Table 2 ?
7. What are the precise meaning of running time and prediction time in the paper? In Abstract, authors stated the proposed framework can reduce running time to second and prediction time to milliseconds. In Section 3.4, authors said the method always recommend models in 3 seconds.  Is the running time the amount of Time the system take to determine the best pipeline? Authors highlight training time is still needed for the prediction. How much time authors give to the best pipeline for training?
8. No real-time applications are demonstrated in the experiments. When the framework only take  3 seconds to return the best pipeline for a dataset, it just means that the search time is short enough. The real-time is misleading readers that authors tackle the real-time applications with AutoML. Their experiments are not strong enough to support the "real-time".
9. The time measurement is not fair for baselines. Authors  ignore the pre-processing time. The running time for predicting a ML pipeline is short from the framework. However, if Auto-Sklearn does not count the time to build surrogate model for Bayesian optimization, Auto-Sklearn can also predict a pipeline in a short time. Authors should clarify how they measure time for predicting a pipeline.

---

### Official Review · AnonReviewer4 · 2020-10-30
**A system to recommend AutoML pipelines for new supervised learning tasks. A practical solution, but need more theoretical discussion and experimental analysis.**

**Rating:** 4
**Confidence:** 4

**Review:**

The paper proposes an efficient way to automatically choose the best or most suitable pipeline for different datasets. The proposed method can accelerate the AutoML using a pre-trained meta module. In particular, the AutoML job of a new supervised learning task can be accomplished without model evaluations, namely zero-shot / real-time AutoML. The meta module is constructed as a graph structure in which each node represents a dataset used for meta-training.

Pros:

1. The target problem in this paper is practical, and the proposed model, GAT is simple and seems effective.

2. The model is built on top of existing AutoML systems and only needs to predict the most suitable pipeline chosen among the existing AutoML systems.

3. The authors claim that this procedure can be done in 3 seconds, i.e., real-time while the other pipelines require 1 minute to train the comparative performance.

Cons:

1. The feature extractor method is based on a Github project, i.e., BYU. However, the Github project doesn’t provide any text description of the technique. To be a self-contained paper, the authors should clarify the details of the method, especially in this paper where the feature is a key input to the GAT. Thus, a more detailed elaboration on how to extract the feature is recommended.

2. The proposed work lacks theoretical or empirical support on why the current feature of the datasets can indicate the choice of the pipeline. For example, given two supervised learning tasks with high dimension features and a limited number of instances, the similarity on two datasets cannot directly indicate using the same pipeline.

3. The proposed method is a systematic solution for AutoML rather than an AutoML method. The paper’s solution leverages multiple existing AutoML pipelines, including OBOD, AutoSklearn, AlphaD3M, and TPOT, and then recommend a pipeline for a new coming supervised learning task by measuring the similarity between the new task’s dataset and all seen dataset. To be specific and avoid misguiding, I’d suggest to name it “Selecting AutoML pipeline”.

4. Compare to other AutoML methods, the proposed dataset embedding-based AutoML solution highly relies on how the new task’s dataset is linked to the existing meta-training datasets. This assumption is difficult to be satisfied in real-world applications.

5. The experimental analysis of this paper is weak. The dataset graph should be visualized with colored points for each pipeline.

6. The description of the dataset is also a key feature to calculate the similarity of two supervised learning tasks. In real work applications, the dataset description is not always well-written in natural language. The use of a dataset description does not always benefit; sometimes, it will be harmful to measuring the similarity of datasets.


Below are some questions:

1. In section 3.1, the authors claim the pipeline embedding is derived by a transformer and form a 1024-dimensional embedding. Can the authors elaborate more details on the inputs of the transformer? Is the representation semantically meaningful?

2. Why the proposed AutoML system can infer faster than only using either one of the pipelines as in Table 2? Do the 3 seconds
only account for the procedure of choosing the best pipeline options?

3. The authors claim this is a real-time autoML system. I am curious about if the system can handle the query in parallel? For example, there are multiple nodes added into the graph at the same time, can the proposed handle this case and how to build the graph for multiple new nodes, and how is the performance?

4. In the current system, the authors assume a fixed number of AutoML systems, i.e., OBOE, AutoSklearn, TPOT, AlphaD3M, Random Forest. Why these five systems selected? What if the user wants to add more systems? Are there any selection criteria for the base systems?

5. I am curious to see some analysis / visualizations on how the current system chooses the best pipeline and which pipeline is typically suitable for datasets of what kinds of types? Maybe a few examples are sufficient to give an intuitive sense.

6. To support the reproducibility of the paper, do the authors have any plan to open source or release the experimental source codes?

---

### Official Review · AnonReviewer2 · 2020-10-30
**Clearly written but low impact, low novelty and a few technical issues.**

**Rating:** 2
**Confidence:** 5

**Review:**


--Summary--
This paper introduces a method for matching an ML pipeline to a dataset using the text features of the pipeline of and the dataset along with some additional metadata about the features of the data. They evaluate and train their model on ~20 datasets and ~5 pipelines.

--Strengths--
The authors introduce an approach to pick a training pipeline for a dataset. I like the fact that the authors provided useful illustrations and clear notation.

--Weaknesses--
This paper suffers from several issues:

**Low impact**
The claim of real-time AutoML is exaggerated:
a) The proposed approach does not work for general AutoML tasks and search spaces such as architecture search, swapping pipeline pieces in and out. Instead it matches datasets to a handful of pre-existing pipelines (~5).

b) Training the model can severely dominate the suggestion time for the proposed algorithm.  This approach focuses on the speed of the first suggestion (zero-shot setting).

c) How do you recover if you make a bad decision about a dataset/algo? The proposed algorithm is not iterative.

d) Are text features sufficient? What about datasets and pipelines which sound similar but are totally different.

**Low novelty**
a) This is a method for matching a dataset to a training pipeline largely using text features from their descriptions/documentation. This is a common information retrieval task.

b) It’s unclear why a GNN and a transformer is needed for a simple feature matching task. Standard retrieval and ranking algorithms such as vector dot product, TF-IDF or LSA could have sufficed.

c) Edges in the GNN are chosen using a kNN threshold applied to the node representations. No structure learning is performed.

**Weak results**
a )The number of datasets (around 20) and pipelines (around 5)  that this approach is evaluated on is woefully inadequate. The results could easily suffer from overfitting.

b) For some datasets test performance is already at 100% e.g. Cardi, Wallrobot, Mofn. These seem like easy tasks. More complex and difficult datasets are needed.

c) No comparison with other methods. e.g. simple information retreival method such as vector dot product, tf-idf or LSA.

**Technical issues:**
a) Description of training setup is lacking. Is it training on the same set of datasets? If so there’s leak of training -> test data.

b) What is the total training time? If it dominates the suggestion. Then this method is not useful since compute has already been spent on fully training the datasets.

---What can make this paper better?---
- The core approach fundamentally needs improvement. A simple pipeline to dataset matching is unlikely to have a significant contribution to science or application.

-  The authors need much higher quality results
a) Many more datasets

b) Difficult datasets: Not ones that achieve close to 100% accuracy.

c) Many more pipelines or complexity of pipeline components.

- To claim Real-time AutoML the authors need to demonstrate that the method works

a) For general AutoML tasks and search spaces such as architecture search, swapping pipeline pieces in and out.

b) The method is iterative in nature. Otherwise it is impossible to recover from poor choices of pipelines.

c) The real-time claim implies an “online” setting, wherein results are fed back and suggestions are offered in real time. The proposed approach does not have the ability to learn on the fly. The current GNN and Transformer representations will be costly to re-train.

---

### Official Review · AnonReviewer1 · 2020-11-03
**Interesting idea that appears to need further work**

**Rating:** 4
**Confidence:** 4

**Review:**

The problem that the authors attempt to solve is to determine what ML pipeline will perform best on any new dataset, without incurring in the extra cost of actually running a large number of such pipelines, as is typically done in AutoML algorithms. The way this paper tackles the problem is to train a neural network that given a new dataset as input, will output a pipeline that is predicted to perform well on that dataset. This neural network is trained on other datasets, for which high performing pipelines are already known. Predicting a pipeline for a new dataset thus only require a forward pass through their NN.

The core details of the solution describes are the following:
* To get labels (in the form of well performing pipelines) for the training set, each dataset in the training set is run through 4 existing AutoML solutions, and the best performing pipeline among those generated this way is selected and used as a ground truth label
* They generate an embedding for each dataset, that combines metafeatures and a textual describtion of the dataset
* As an additional step, the embedding describing the dataset is transformed into a graph in the following way: all existing datasets (during prediction including the new dataset) are represented as a graph, where each node corresponds to a dataset and is connected to the closest k graphs according to the distance of the embeddings
* The above graph is then passed as input to a GNN, to predict the final pipeline
* In addition, each node has a feature describing the selected pipeline, with the value set to 0 for the current active node that is being passed as the input dataset, as far as I can tell this is the only way in which the NN is aware what the current active node is.
* The problem is framed as a multiclass classification problem with a discrete set of possible pipelines to choose from, thus the final softmax layer of the network simply outputs a probability distribution over this discrete set of pipelines


The idea is interesting, although it is not really clear why the exact proposed architecture would be the best way to approach the issue. From a theoretical point of view it's not clear why a simpler architecture would not work better, or as at least give comparable performance. Even more importantly, the experimental results leave some doubts about the effectiveness of the method.

I found the exposition also hard to follow in places. More precisely, a lot of the discussion of the use of GNNs in the introduction is hard to follow without the context of how the GNN is used exactly, which is explained only later in the text. The second paragraph of the introduction, about the use of GNNs, is very hard to understand without first knowing what exactly is passed in as input and output of the NN. Same for the forth paragraph. I would suggest to leave a lot of these discussions out or put them later in the paper. More in general the proposed architecture is not really clear until the reader reads all the details of the paper, a better and more self-contained introduction would help the comprehension of the paper.

The biggest concern for me however are the results, in particular how the method, although relatively complex, seems to perform no better (or only very slightly better) than random forests with default parameters, as seen in the results of Table 2, and Figure 2 of the Supplemental material. In addition the authors seem to have tested many different versions of the algorithm, as per the supplemental material. This in itself is positive and I appreciate the ablation studies. The issue is that even if all variations would all perform no better than the random forest baseline, some architectures would appear to perform better than it just due to noise. Making the result even more clear cut.

Overall I believe the approach of the authors is interesting, but further work seems necessary to reach a publishable paper, both from an exposition point of view, and from an experimental one. Unfortunately the paper misses strong and compelling evidence that the method really gives results than simple baselines.

#### Additional comments / Questions

When describing embeddings: the following sentence is not very clear unless one already is familiar with the relevant literature, in which case it does not add anything: ". An unsupervised corpus of text is transformed into a
supervised dataset by defining content-target pairs along the entire text: for example, target words
that appear in each sentence, or target sentences which appear in each paragraph"

It would be good to have the code available, as quite a few low level details are left out of the paper.

How big is the variety of pipelines selected? This would be interesting to know.

In "Embeddings": corpra -> corpora

In "Methods": their relationships -> it is not really clear what this means in this contest

---

### Official Review · AnonReviewer5 · 2020-11-05
**Interesting, unique approach to meta-learning for AutoML leveraging deep learning; empirical performance somewhat underwhelming**

**Rating:** 4
**Confidence:** 4

**Review:**

This paper presents a very interesting idea of utilizing the documentation for the data and the operators in the pipeline to generate meta-features for meta-learning. This is a very novel application of graph neural networks GNNs and language models for AutoML meta-learning. This view of meta-learning takes a very intuitive on a very high level. The use of the outputs of existing AutoML systems (such as auto-sklearn, TPOT, etc) is also very intuitive and well motivated. All these intuitive ideas are put together into a novel AutoML recommendation architecture making use of modern deep learning components.


However, in its current form, I am somewhat unsatisfied with the paper. Firstly, even with all this sophisticated meta-learning, it appears that a RandomForest is fairly competitive at 3 seconds (5 wins, 4 ties, 9 losses to zero-shot) and at 1 minutes (5 wins, 8 ties, 5 losses to zero-shot), which is somewhat disappointing. Given that XGBoost and LightGBM are usually more efficient and accurate, chances are, they would improve upon RandomForests. It seems that the gain from the proposed sophisticated meta-learning is not fully realized with the empirical evaluation.


Beyond this, there are various choices made in the empirical evaluation which is not very well justified, and various technical details that are missing in the paper:

- Why are we focusing on 3 seconds, why not 1 second or 1 minute? Also, does 3 seconds constitute just the recommendation of pipelines or the recommendation + training. Different parts of the text imply different things -- End of section 3 says "Our method can always recommend a model in 3 seconds, but training is still needed for prediction". Then in the beginning of section 4, the authors say "Our new zero-shot AutoML approach is the only AutoML system that provides predictions within 3 seconds".
- In the introduction, the authors say "... runs the pipeline and tunes its hyper-parameters within three seconds", but nowhere in the text could I precisely find what happens once the system recommends a pipeline -- is it just trained as it is or is hyper-parameter optimization or CASH run on it?
- Is the generation of the meta-data (document parsing, statistics generation), which can take significant amount of time for decent sized data sets, part of the 3 seconds?
- For each node during meta-learning (training of the GNN and the fusion networks), the potential pipelines  $\mathcal{P}_{\mathcal{C}} ( \mathcal{D} )$ are the ones generated by auto-sklearn, TPOT, OBOE, etc, on $\mathcal{D}$ and "the output of the GAT is a probability distribution over pipelines for each node".
  - What is the set of potential pipelines over which the output probability distribution is defined for a new test data set $\mathcal{D}{\mbox{new}}$ for which we have no potential pipelines $\mathcal{P}_{\mathcal{C}} ( \mathcal{D}{\mbox{new}} )$. This is not clear from the description and is a very important part of the "zero-shot" learning.

---

### Author Response · Authors · 2020-11-24
**Our response**

We thank the reviewers for their time and are working on an improved system and paper.

1. Transformers: this work is the first to use Transformers in AutoML, as a follow-up to the workshop paper on AutoML language embeddings (2019) and symposium paper on Zero-Shot AutoML (2020).

2. Privileged AutoML: Current AutoML systems use the data, without the description of the data, without the description of the machine learning primitives, and without using results of other AutoML systems. The current AutoML input is limited. We propose to use privileged information for training: descriptions of the data and the primitives and results of other AutoML systems. Transfer learning by training our system on data + {dataset descriptions, primitive descriptions, and other AutoML systems} to a system receiving only the data as input, yields far better results than any of the existing systems given the same time, as shown in the supplementary material. In our follow-up work we will compare our privileged AutoML with other systems without any privileged information.

3. Stronger baselines: we are extending our system to use gradient boosting algorithms that perform well on tabular data classification.

4. Diverse AutoML systems: we are extending our system to training using additional AutoML systems: H2O and AutoGluon.

5. Time scale: We focused on a 3-seconds time-frame which is around the upper bound that would allow user interaction with such an AutoML system. The system may be run in parallel by adding datasets in random order. Our system performs in 3-seconds as well as other methods given 1 minute. In a follow-up we show improved performance given the same time.

6. Hyper-parameters: Our approach recommends pipelines that are originally generated by other AutoML systems. As a result, these pipelines include tuned hyper-parameters. While our meta-learning algorithm does not perform hyper-parameter tuning by itself, the predicted pipelines include tuned hyper-parameters since they originate from other tools.

7. Pipeline distribution and diversity: The predicted pipeline distribution is over the neighbors of the relevant dataset. We plot the normalized difference in cross validation accuracy between our system and baselines split by primitives used demonstrating better performance across diverse primitives.

8. Writing: we are re-writing the paper based on transfer learning from a privileged AutoML system (teacher) to a base AutoML system (student).

9. Code: We have a new Github repo which is robust and well documented, which will be released upon publication of a new paper.

10. Training: we also measure performance by k-fold cross-validation.

11. Datasets: we are extending our data to a total of 1000 datasets, with 800 datasets for training, 100 for validation, and 100 for testing. In comparison current AutoML benchmarks use around 30 test datasets: https://openml.github.io/automlbenchmark/benchmark_datasets.html

---

### Decision · Program_Chairs · 2021-01-07
**Final Decision**

**Decision:**

Reject

**Comment:**

The paper presents an algorithm for real-time auto-ML based on zero-shot learning, which matches an ML pipeline to a dataset via the meta-features of the pipeline and the dataset.  It aims to address an important problem, and the idea of the proposed solution seems interesting. However, there are several issues with the current draft:

(1) A central question is how to justify the complexity of the proposed solution, given that simple alternatives, such as Random Forest, can achieve similar or better results than the proposed method.

(2) The proposed solution lacks in novelty contribution of methodology and theoretical justification

(3) Various technical details are missing in the current draft.

The authors' feedback did not fully address the issues above. We hope that reviews can help the authors improve the draft for a strong publication in the future.